# Unsupervised Spatiotemporal Data Inpainting

## Abstract

We tackle the problem of inpainting occluded area in spatiotemporal sequences, such as cloud occluded satellite observations, in an unsupervised manner. We place ourselves in the setting where there is neither access to paired nor unpaired training data. We consider several cases in which the underlying information of the observed sequence in certain areas is lost through an observation operator. In this case, the only available information is provided by the observation of the sequence, the nature of the measurement process and its associated statistics. We propose an unsupervised-learning framework to retrieve the most probable sequence using a generative adversarial network. We demonstrate the capacity of our model to exhibit strong reconstruction capacity on several video datasets such as satellite sequences or natural videos.

## 1 Introduction

We consider the problem of reconstructing missing information from image sequences. The problem occurs in many different settings and for different types of sequences. For example, in remote sensing applications, satellite imagery are frequently occluded by meteorological perturbations such as clouds and rains (Singh & Komodakis, 2018). Recovering missing satellite data is an active research topic. Approaches range from simple interpolation to sophisticated data assimilation methods. The latter is often a model-based approach that relies on analytical models of the underlying observed phenomenon (Ubelmann et al., 2015; Sirjacobs et al., 2011; Lguensat et al., 2017). Model-free data based methods have also been developed such as DINEOF (Alvera-Azcárate, 2011). Note that for physical observation modeling problems of this type, there is never any direct supervision available. Another example concerns natural videos. Here, information can be occluded by moving objects such as fences (Yamashita et al., 2010), raindrops (Qian et al., 2018), persons (Kim et al., 2019), stains on photographic films (Tang et al., 2011). Video and image imputation have given rise to a large body of literature. Recent Deep Learning (DL) advances have motivated the development of general imputation methods relying on generative models such as GANs (Wang et al., 2018a; Xu et al., 2019; Kim et al., 2019). They all make use of supervision and require the availability of a ground truth, which is absent in many real-world problems. Data driven supervised methods have thus attained impressive results and are able to accurately complete a large missing region. However, reconstructing the missing information in videos when supervision is unavailable is still an open problem and there have been only a few works exploring this direction. For example, Newson et al. (2014) propose a simple but effective method for occlusions in natural videos that replaces occluded parts with information from their neighborhood.

We consider here unsupervised video reconstruction. We propose a model which can be used on different types of image sequences, physical or natural videos, and for a large variety of occlusion processes. Our method does not make any assumption on the nature of the image sequence, it does not require any prior knowledge like most methods used for physical images do. It is especially well suited when the occlusion is complex thus forbidding the use of ad hoc techniques, e.g., the patch method of Newson et al. (2014). The method extends to sequences from ideas recently developed for still images based on generative networks (Bora et al., 2018; Pajot et al., 2019; Li et al., 2019). This is up to our knowledge the first attempt to solve the problem of unsupervised video completion using general ML methods. This method is fully data driven and does not use any hand-defined analytical prior on the signal. Priors on the unobserved signal are directly learned from the data for solving an underlying inverse problem. The method is then applicable to a large variety of video signals.

Our main contributions are the following:

- We propose a new framework and model for large-scale image sequence inpainting learning, in a fully unsupervised context.
- This model can be used for a variety of image sequences and for different occlusion processes.
- Extensive evaluations are performed on realistic simulated satellite data and on natural videos with different occlusion processes.

## 2 METHOD

### 2.1 PROBLEM SETTING

We suppose that there exists an unknown spatiotemporal sequence $\boldsymbol{x} \sim p_{\mathbf{X}}, \boldsymbol{x} \in \mathbb{R}^{C \times T \times H \times W}$, where $\boldsymbol{x}$ is a tensor denoting a $C$-channel sequence composed of $T$ frames of $H \times W$ pixels. We denote $\boldsymbol{x}_t$ the $t$-th frame of the sequence and $\boldsymbol{x}_{t_1}^{t_2}$ the subsequence from the $t_1$-th to the $t_2$-th frame inclusive. With this notation, $\boldsymbol{x} = \boldsymbol{x}_1^T$. We do not have access to the original signal $\boldsymbol{x}$ but only to corrupted observation sequences of this signal $\boldsymbol{y} \sim p_{\mathbf{Y}}, \boldsymbol{y} \in \mathbb{R}^{C \times T \times H \times W}$. Our objective is to reconstruct $\boldsymbol{x}$ from the corresponding observation $\boldsymbol{y}$. For example, $\boldsymbol{x}$ can be sea surface temperature (SST) at successive times while image sequence $\boldsymbol{y}$ is SST measurements via IR satellites occluded by moving clouds. We will suppose that $\boldsymbol{y}$ is obtained from $\boldsymbol{x}$ via a measurement process modeled through a stochastic operator $F$ as follows:

$$\boldsymbol{y} = F(\boldsymbol{x}, \boldsymbol{m}) = \boldsymbol{x} \odot \boldsymbol{m} + c \cdot \bar{\boldsymbol{m}} \tag{1}$$

where $\boldsymbol{m} \sim p_{\mathbf{M}}$ is an occlusion mask, generated from a known distribution with the same size as $\boldsymbol{x}$ and with components in $\{0, 1\}$, where 0 holds for a masked pixel. $\bar{\boldsymbol{m}}$ denotes the complement of $\boldsymbol{m}$, $\odot$ is the element-wise multiplication, all the masked pixels are supposed to be reset to a constant $c$ which could be 0 or 1 depending on the observation process (see Section 3). Random variables $\mathbf{X}$ and $\mathbf{M}$ are assumed to be independent and $F$ is assumed differentiable w.r.t. $\boldsymbol{x}$. In the following, we will suppose that one can retrieve the mask $\boldsymbol{m}$ directly from the observation $\boldsymbol{y}$. This is not very restrictive since in most situations this is easy to do. We denote $T$ the mask extractor $T(\boldsymbol{y}) = \boldsymbol{m}$.

Our objective is then to recover the sequence $\boldsymbol{x}$ from the observations $\boldsymbol{y}$ and the corresponding binary masks $\boldsymbol{m}$. Adopting a probabilistic viewpoint, we want to select a reconstruction $\boldsymbol{x}^*$ which is the most plausible under the posterior distribution $p_{\mathbf{X}|\mathbf{Y}}(\cdot|\boldsymbol{y})$.

### 2.2 MODEL

We formulate the problem as finding the most probable sequence conditioned on observations:

$$\boldsymbol{x}^* = \arg\max_{\boldsymbol{x}} \log p_{\mathbf{X}|\mathbf{Y}}(\boldsymbol{x}|\boldsymbol{y}) = \arg\max_{\boldsymbol{x}} \log p_{\mathbf{X}}(\boldsymbol{x}) + \log p_{\mathbf{Y}|\mathbf{X}}(\boldsymbol{y}|\boldsymbol{x}) \tag{2}$$

The prior term $\log p_{\mathbf{X}}(\boldsymbol{x})$ is unknown since we are in an unsupervised setting, while the likelihood $\log p_{\mathbf{Y}|\mathbf{X}}(\boldsymbol{y}|\boldsymbol{x})$ does not lead to analytical or simple computational solution.

To tackle these issues, let us introduce a mapping $G : \mathbf{Y} \mapsto \mathbf{X}$, parameterized by a neural network $\varphi$ and associating measurement $\boldsymbol{y}$ to its estimate $\boldsymbol{x}$. $G$ will allow us to approximate the underlying distribution of training sequences. By plugging $G(\boldsymbol{y})$ into Equation 2, the objective becomes:

$$G^* = \arg\max_{G} \underbrace{\mathbb{E}_{\boldsymbol{y} \sim p_{\mathbf{Y}}}[\log p_{\mathbf{X}}(G(\boldsymbol{y}))]}_{\text{prior}} + \underbrace{\mathbb{E}_{\boldsymbol{y} \sim p_{\mathbf{Y}}}[\log p_{\mathbf{Y}|\mathbf{X}}(\boldsymbol{y}|G(\boldsymbol{y}))]}_{\text{likelihood}} \tag{3}$$

### 2.3 PRIOR HANDLING

Let us first handle the prior term in Equation 3. We want the distribution induced from $G(\boldsymbol{y})$ to be close to $p_{\mathbf{X}}$. In order to do so, we will use an adversarial approach. We will build on the ideas introduced in Bora et al. (2018); Pajot et al. (2019) for still images. The process is illustrated in Figure 1. For a given observation $\boldsymbol{y}$, we want to generate an approximation of the unknown true sequence $\hat{\boldsymbol{x}} \equiv G(\boldsymbol{y})$. The prior $p_{\mathbf{X}}$ being unknown, the only available information source is the observation $\boldsymbol{y}$ and the noise prior $p_{\mathbf{M}}$. For a given generated signal $\hat{\boldsymbol{x}}$, we compute a corrupted version of $\hat{\boldsymbol{x}}$ through

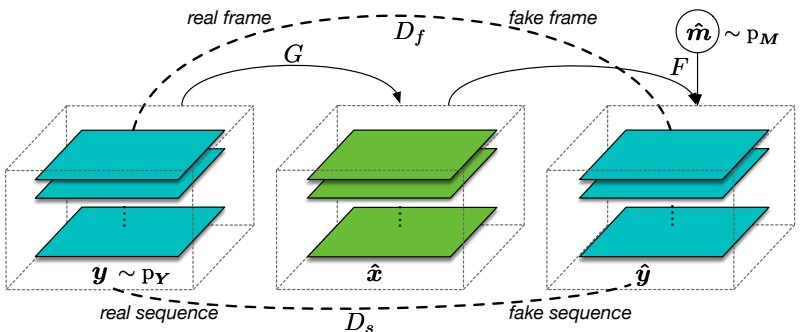

Figure 1: Schema of our model. Generator $G$ takes a sequence $\boldsymbol{y}$ and outputs an inpainted sequence $\hat{\boldsymbol{x}}$; measurement process $F$ takes the inpainted sequence then outputs fake observations $\hat{\boldsymbol{y}}$.

the known mask $\hat{\boldsymbol{m}}$, $\hat{\boldsymbol{y}} \equiv F(\hat{\boldsymbol{x}}, \hat{\boldsymbol{m}})$ with $\hat{\boldsymbol{m}} \sim \mathrm{p_M}$. We will train $G$ to make the distributions of $\boldsymbol{y}$ and $\hat{\boldsymbol{y}}$ indistinguishable. In order to succeed, the generator $G$ will have to remove the corruption from $\boldsymbol{y}$ and recover a sample $\hat{\boldsymbol{x}}$ from distribution $\mathrm{p_X}$. Generator $G$ will then act as an inpainter conditioned on $\boldsymbol{y}$. This will enforce the distribution of the reconstructed sequences $\hat{\boldsymbol{x}}$ to be close to the distribution of true ones $\boldsymbol{x}$ and maximize the prior term.

A direct application of the adversarial training idea suggests using a discriminator operating directly on the sequences. We found out that using an additional discriminator on frames worked better than using a unique one operating on sequences. We then use two discriminators $D_s$ and $D_f$ respectively associated with whole sequences and with individual frames to optimize $G$. $D_s$ separates sequences $\boldsymbol{y}$ and $\hat{\boldsymbol{y}}$. $D_f$ distinguishes real frames $\boldsymbol{y}_t$ from fake ones $\hat{\boldsymbol{y}}_t$. The loss function used for training $G, D_s$, and $D_f$ is:

$$\min_G \mathcal{L}(G) = \max_{D_s, D_f} \mathbb{E}_{\boldsymbol{y} \sim \mathrm{p_Y}, \hat{\boldsymbol{y}} \sim \mathrm{p_Y^G}}[\log D_s(\boldsymbol{y}) + \log(1 - D_s(\hat{\boldsymbol{y}})) + \frac{1}{T}\sum_{t=1}^{T} \log D_f(\boldsymbol{y}_t) + \log(1 - D_f(\hat{\boldsymbol{y}}_t))]$$

(4)

with $\mathrm{p_Y^G}(\boldsymbol{y}) \equiv \mathbb{E}_{\boldsymbol{m} \sim \mathrm{p_M}, \boldsymbol{x} \sim \mathrm{p_X^G}}[\mathrm{p_{Y|X,M}}(\boldsymbol{y}|\boldsymbol{x}, \boldsymbol{m})]$, corresponding to the distribution of the corrupted sequences $\hat{\boldsymbol{y}}$ generated via the measurement operator $F$. $\mathrm{p_X^G}(\boldsymbol{x})$ is the distribution of $\hat{\boldsymbol{x}}$ induced by $G$ from $\boldsymbol{y}$, i.e. $\hat{\boldsymbol{x}} = G(\boldsymbol{y})$.

## 2.4 LIKELIHOOD HANDLING

Let us now handle the likelihood term in Equation 3:

$$\mathbb{E}_{\boldsymbol{y} \sim \mathrm{p_Y}}[\log \mathrm{p_{Y|X}}(\boldsymbol{y}|G(\boldsymbol{y}))].$$

(5)

This likelihood is maximised when we are able to perfectly reconstruct $\boldsymbol{y}$ from $G(\boldsymbol{y})$. One way to ensure this property is to constrain $G$ to directly use $\boldsymbol{y}$ for the non occluded area of the reconstructed image $G(\boldsymbol{y})$. This can be easily achieved through the following mapping:

$$G(\boldsymbol{y}) \equiv \varphi(\boldsymbol{y}) \odot \bar{\boldsymbol{m}} + \boldsymbol{y} \odot \boldsymbol{m}$$

(6)

where $\varphi$ is a NN responsible for reconstructing the missing part of $\boldsymbol{y}$, $\boldsymbol{m} = T(\boldsymbol{y})$ is the mask retrieved from $\boldsymbol{y}$. $G$ maps $\mathbf{Y}$ to $\mathbf{X}$ with the help of mask $\boldsymbol{m}$ to ensure that the network will only generate values for occluded pixel, while keeping all the information from $\boldsymbol{y}$. To summarize, optimizing the prior term amounts at training $\varphi$ for inputting the missing pixels while optimizing the likelihood term is simply achieved by copying the non occluded portion of $\boldsymbol{y}$.

## 2.5 TRAINING

$G$ is optimized by descending the prior loss and $D_s, D_f$ by ascending it. Sequence discriminator $D_s$ focuses on temporal dependence and coherence of pixel changes. Frame discriminator $D_f$ keeps an eye on spatial feature coherence of observation frames.

## 3 Experiments

We evaluate our model on four datasets, characteristic of different types of image sequences. The first one, SST, is a realistic simulation of satellite observations. The other three are natural video datasets: FaceForensics++, KTH, and BAIR, initially respectively used as benchmarks for forgery detection, motion detection, and video prediction.

### 3.1 Datasets

**SST** The Sea Surface Temperature dataset used for the experiments includes 2 subsets of GLOBAL Sea Physical Analysis and Forecasting Product[1] from E.U. Copernicus Marine Service Information. This is a monitor system providing simulated but realistic global ocean SST data, which integrates satellite-derived and *in situ* data by assimilation. Our dataset is a part of the hourly mean SST, the finest timescale we have access to. The data we use is a part of the archive of analysis integrating real-world data. We retrieved our training-and-validation set and test set respectively from two different marine regions. Detailed data description and information for accessing the dataset are provided in Appendix A.

**FaceForensics++ (Rössler et al., 2019)** This dataset contains 1000 videos of non-occluded face movements on a static background. It was initially created for forgery detection. In our case, we extracted the faces from the original unforged videos with `face_recognition`[2], thus keeping only the changing component of the videos. The faces have been cropped and resized to 64×64.

**KTH (Schuldt et al., 2004)** A human action dataset containing 2391 video clips of 6 human actions. The videos have been recorded with 25 subjects in different environments. All frames have been resized to 64×64.

**BAIR Robot Pushing Dataset (Ebert et al., 2017)** This dataset contains 44374 videos recorded by an one-armed robot. It pushes objects and changes movement direction in a stochastic manner. All videos share similar tabletop with static background. All frames have been resized to 64×64.

### 3.2 Measurement Processes

The above datasets provide ground truth videos without corruption. In order to generate corrupted observation sequences, we simulate different types of occlusion depending on the nature of the videos. Each corruption process is defined as a stochastic operator $F$ as in Equation 1 with mask distribution $p_M$. For a given video one then generates a sequence of random masks, one mask being then associated to each frame of the sequence. Note that except for the Remove-Pixel corruption process where two successive corruptions are independent, for all processes, the generated corruption sequences are time-dependent: the corruption pattern at time $t$ will depend on the one at time $t-1$.

**Cloud** This process is specific for the SST dataset. It simulates realistically video cloud masks on satellite images. Cloud masks are simulated using Liquid Water Path (LWP) data (measured in g/m$^2$), which characterizes the total amount of liquid water present in the atmosphere between two points. The LWP data are generated by PyCLES (Pressel et al., 2015)[3], a large eddy simulation system. It simulates the evolution of clouds in time based on a variant of anelastic equations of atmospheric motion. Collected LWP data record mask videos of clouds. The binary masks are then obtained by setting the image pixels to 0 when their LWP value is above a threshold. This produces realistic cloud coverage of the captured regions, see Figure 2a. Pixels occluded by the mask are set to $c = 1$. Thresholds are selected in the interval 55 to 80 g/m$^2$ to simulate clouds at different occlusion rates. Statistics about the occluded area at different thresholds are presented in Table 2a. For simulating occlusion, for each SST image sequence, we sample randomly a sequence of masks from the LWP dataset to be applied to the SST sequence.

---

[1] http://marine.copernicus.eu/services-portfolio/access-to-products/
[2] https://github.com/ageitgey/face_recognition
[3] https://github.com/pressel/pycles

**Raindrops**  This process is a simplified model of random raindrops between subject and camera, taking into account a blurring effect when raindrops leave traces during exposure. It generates a set of white bars, each with a random length $\theta_l$ and a constant width $w$. Bars move down at a random speed $\theta_v$, starting from a random initial position $\theta_p$. All these values are normalized w.r.t the frame edge length in $]0, 1[$. The number of raindrops is pre-defined. Bars return to the top once completely out of frame, see Figure 2. Pixels occluded by the mask are reset to $c = 1$. Note that as for Cloud, this is a time-dependent measurement process, meaning that two successive masks are correlated.

**Remove-Pixel**  This measurement roughly mimics severe damages on vintage films. It masks randomly a fixed proportion $p \in \; ]0, 1[$ of pixels at each time step and reset them to $c = 0$, see Figure 2. Mask for each frame is generated independently regardless the evolution of time. This is the only time-independent measurement considered here.

**Moving-Vertical-Bar**  This simple measurement operator generates a vertical white bar crossing the sequence, very roughly mimicking a fence or any similar obstacle. The bar is generated with the following distribution parameters: width $\theta_w$, initial position $\theta_p$, horizontal constant velocity $\theta_v$. These values are in $]0, 1[$ as for Raindrops. The moving direction is chosen randomly. The bar reappears on the opposite side once it reaches the border. Masked pixels in observation are reset to $c = 1$. This is a time-dependent measurement.

### 3.3 BASELINES

**Unsupervised Approaches**  We use two unsupervised baselines, one adapted for SST and the other one specific of natural videos. The former is DINEOF (Alvera-Azcárate, 2011). This is a state-of-the-art data-driven completion method in geophysics, and it has been used for SST observations, chlorophyll, salinity etc. It is a parameter-free interpolation technique based on empirical orthogonal function (EOF). It adopts an iterative algorithm that calculates at each iteration a truncated decomposition of EOF from known pixels, then replaces the values marked as missing by a reconstruction from calculated EOF. DINEOF does not make any assumption on the form of missing area and as such could be used for other domains as well and for different types of complex occlusion processes. However, DINEOF has been developed for remote sensing and does not ensure the coherence between different input channels (e.g. for RGB images).

The other one is Newson et al. (2014), one of the very few methods for unsupervised natural video inpainting. It is representative of patch-based approaches and it is still today state-of-the-art for many natural video occlusion processes. It searches for the nearest neighbors of occluded area using an Approximate Nearest neighbor (ANN) search with PatchMatch (Barnes et al., 2009) finding for each occluded pixel the best corresponding non-occluded patch. The occluded area is then reconstructed by assembling information from these neighbors at multiple scales. The form of the researched patches is supposed to be rectangular cuboids, e.g. a $5 \times 5 \times 5$ spatiotemporal tensor, which limits its capability to adapt to more complex cases like Cloud, Raindrops, or Remove-Pixel.

**Supervised Approaches**  As already mentioned, there exists several supervised approaches to sequence inpainting (Huang et al., 2016; Xu et al., 2019; Kim et al., 2019). In order to evaluate the performance of our unsupervised method w.r.t supervised ones, we compared with two supervised baselines. As our goal is not to beat state-of-the-art supervised techniques, we used two supervised adaptation of our model, respectively trained using unpaired and paired supervision. They are described below.

UNPAIRED VARIANT  This is a supervised variant of our model in which we have access to unpaired samples from $p_{\mathbf{X}}$ and $p_{\mathbf{Y}}$. The model is illustrated in Appendix C. Because we have access to clean $\boldsymbol{x}$ data, it is then possible to supervise the approximation $\hat{\boldsymbol{x}} = G(\boldsymbol{y})$ by discriminating directly between samples $\boldsymbol{x}$ from the signal distribution and the output of the reconstruction network $\hat{\boldsymbol{x}}$.

PAIRED VARIANT  Here we have access to corrupted-uncorrupted pairs $(\boldsymbol{y}, \boldsymbol{x})$ from the joint distribution $p_{\mathbf{Y}, \mathbf{X}}$. Given the masked image $\boldsymbol{y}$, the reconstruction is obtained by regressing $\boldsymbol{y}$ to the associated complete image $\boldsymbol{x}$ using a $L^1$ loss. In order to avoid blurry samples, we add an adversarial term in the objective, which helps $G$ to produce realistic samples. This model is similar to the Vid2Vid (Wang et al., 2018b) model, except that they rely on optical flow which is not available in our case because of the masked regions. The model is illustrated in appendix C.

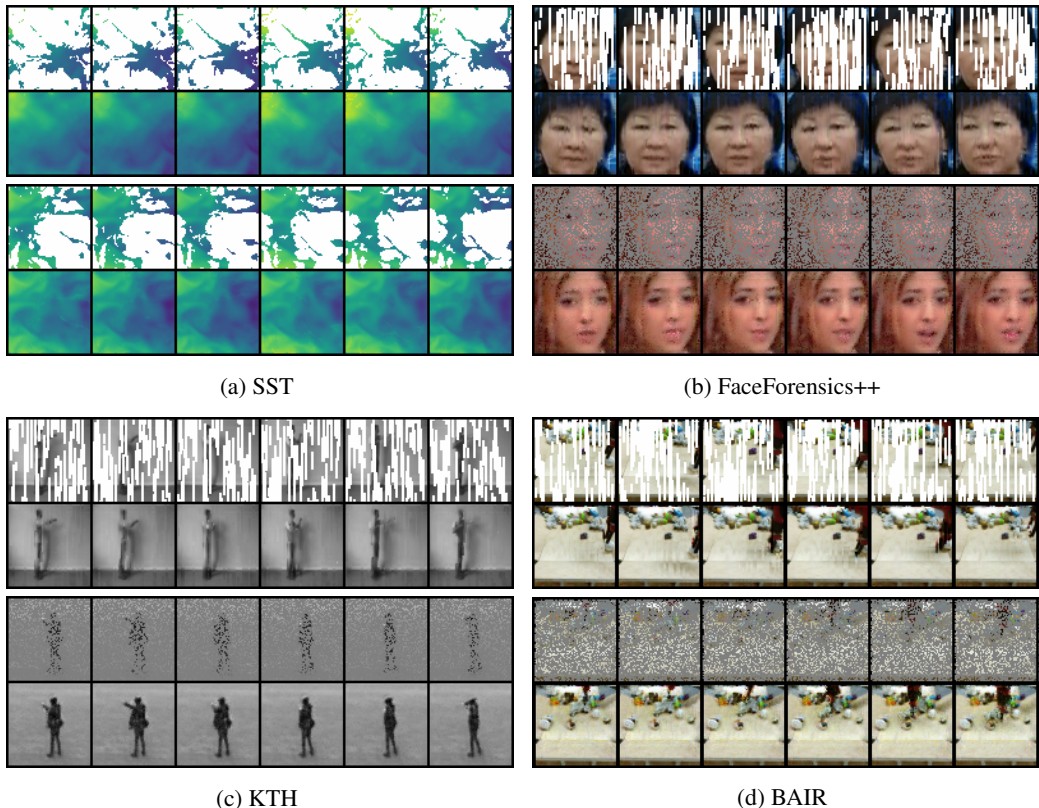

(a) SST

(b) FaceForensics++

(c) KTH

(d) BAIR

Figure 2: Samples from test sets. SST data (a) are masked with Cloud, natural video datasets (b,c,d) are masked with Remove-Pixel and Raindrops. Sequences are accelerated 3 times to make movements more visible. Each sample from top row to bottom: observed $y$, and recovered $\hat{x}$.

### 3.4 NETWORK ARCHITECTURE AND TRAINING DETAILS

We use the same networks for all the experiments. For generator $G$, we use a ResNet-type self-attention network (Zhang et al., 2019), which is composed of 3D-ResNet blocks and spatial self-attention layers. Frame discriminator $D_f$ is a 2D convolutional NN trained for binary classification. Sequence discriminator $D_s$ uses the same structure as $D_f$ but with 3D convolutions. These networks can process sequences of any time length. See Appendix B for more details about the networks.

Let us now detail the training procedure for each dataset: (a) For SST data, the model is trained on 300 sequences, validated on 66 sequences, and tested on 60 sequences. Each sequence is composed of 24 frames. We use SST data degraded by cloud masks at LWP threshold 70 g/m$^2$ for training since they include sufficient information for both SST and cloud dynamics. (b) For FaceForensics++, KTH, and BAIR, we pick randomly 5% data for validation, another 5% data for test, and keep the remaining for training. Sequences are truncated or padded to 35 frames to be able to fit into GPU memory.

For time-independent process Remove-Pixel, we use plain pixel value as feature and directly let the sequence discriminator capture the dynamics. For time-dependent Raindrops, Moving-Vertical-Bar and Cloud, we further reinforce the sequence discriminator $D_s$ to focus on temporal component by extracting inter-frame difference feature since the underlying dynamics reflected by this feature is more expressive than plain pixels. $D_s$ will therefore distinguish between $\phi \equiv [y_2 - y_1, \ldots, y_N - y_{N-1}]$ and $\hat{\phi} \equiv [\hat{y}_2 - \hat{y}_1, \ldots, \hat{y}_N - \hat{y}_{N-1}]$.

We use hinge loss for Equation 4 as in Zhang et al. (2019). Following standard practice, all three networks are trained using Adam optimizer with a learning rate of $1 \times 10^{-4}$ and $(\beta_1, \beta_2) = (0, 0.999)$. All networks are initialized with normal distribution with a gain of 0.02. We apply spectral normalization for all parametric layers. The experiments were made on one NVIDIA GeForce GTX TITAN X GPU. [4]

---

[4]Code and video samples available at: https://sites.google.com/view/unsup-video-inpaiting

| Dataset | Method | Raindrops | | | Remove-Pixel | | | Moving-Vertical-Bar | | |
|---|---|---|---|---|---|---|---|---|---|---|
| | | FID | FVD | MAE | FID | FVD | MAE | FID | FVD | MAE |
| FF++ | Ours | **43.72** | **1574.89** | **.0834±.0187** | **93.28** | **1460.02** | .0894±.0137 | 19.12 | 493.57 | .1304±.0972 |
| | (1) | 75.93 | 3424.11 | .1208±.0272 | 110.15 | 3091.67 | **.0752±.0161** | 56.58 | 5775.25 | .3286±.0815 |
| | (2) | —* | —* | —* | —* | —* | —* | **9.04** | **316.55** | **.0494±.0501** |
| KTH | Ours | **56.56** | **2522.81** | **.0380±.0062** | **56.16** | **2639.24** | .0429±.0037 | 39.05 | 588.94 | .0711±.0505 |
| | (1) | 71.69 | 6400.44 | .0522±.0073 | 82.45 | 6660.02 | **.0403±.0040** | 34.90 | 3408.19 | .0959±.0402 |
| | (2) | —* | —* | —* | —* | —* | —* | **11.88** | **354.01** | **.0268±.0403** |
| BAIR | Ours | **27.33** | **1194.19** | **.0821±.0153** | **53.80** | **2073.90** | **.0997±.0087** | 11.55 | 496.38 | .1619±.0590 |
| | (1) | 89.87 | 4456.08 | .2345±.0274 | 140.20 | 4014.17 | .1424±.0103 | 67.06 | 7361.77 | .5579±.0766 |
| | (2) | —* | —* | —* | —* | —* | —* | **10.31** | **340.97** | **.1082±.0873** |

Table 1: Results for FaceForensics, KTH, and BAIR. Compared with (1) Alvera-Azcárate (2011) and (2) Newson et al. (2014). *Could not terminate.

## 3.5 Evaluation Metrics

Our objective is to find the most plausible sequence. We use as main performance measures of the generated frames, Fréchet Inception Distance (FID, Heusel et al., 2017) and Fréchet Video Distance (FVD, Unterthiner et al., 2018). Both compare the activation distribution of the generated samples from $p_X^G$ to the real one sampled from $p_X$. These distributions are extracted from activation layers of NNs, which are pre-trained respectively on natural image classification tasks for FID and video classification tasks for FVD. The two distances are calculated for the whole sequence including occluded and non-occluded region. Besides FID and FVD, we also evaluate the reconstruction error as a complimentary metric. We use for that Mean Average Error (MAE), which indicates the absolute deviation from the real data. MAE is calculated solely within the occluded area.

## 4 Results

### 4.1 Comparison with Baselines

**Results for SST Data** Table 2a shows the results for SST data with simulated clouds at different occlusion rates. For most occlusion rates, the generated sequences have an MAE under 0.1°C which is well below the reference baseline (see Table 2a). They also have good FID and FVD values, which means that they are spatially and temporally realistic (See Figure 2a for examples). For heavily occluded area, our model can realistically reconstruct the data around the border, the reconstruction near the center of the cloud is of lower quality. We compare our results in Table 2b at 70% occlusion, with DINEOF, the state-of-the-art agnostic method for image sequence reconstruction in IR images. The error reduction w.r.t. DINEOF is about 40% for MAE. We have not been able to obtain results for Newson et al. (2014) because the highly complex Cloud masks make searching for valid cuboid patches in both occluded and non-occluded area extremely hard. Consequently, the algorithm remains blocked in its initialization step and is prevented from the completion. Note that Newson et al. (2014) is specifically designed for inpainting in natural videos so that it is not adapted for this type of data.

| LWP (g/m²) | Occluded Area (%) | FID | FVD | MAE (°C) |
|---|---|---|---|---|
| 55 | 79.9± 9.6 | 32.49 | 134.40 | .1273±.0443 |
| 60 | 69.6±12.8 | 22.95 | 79.13 | .1047±.0396 |
| 65 | 55.9±15.1 | 17.75 | 75.07 | .0988±.0378 |
| 70 | 39.5±14.6 | 8.01 | 40.76 | .0739±.0324 |
| 75 | 24.5±11.5 | 5.58 | 30.07 | .0698±.0305 |
| 80 | 13.4± 7.8 | 1.77 | 9.89 | .0497±.0237 |
| All | 47.1±11.9 | 14.76 | 61.55 | .0874±.0347 |

(a) Results with clouds generated at different LWP thresholds.

| Method | FID | FVD | MAE (°C) |
|---|---|---|---|
| Ours | **8.01** | **40.76** | **.0739±.0324** |
| Alvera-Azcárate (2011) | 27.99 | 323.61 | .1214±.0248 |
| Newson et al. (2014) | —* | —* | —* |

(b) Comparison of results with clouds at LWP threshold 70 g/m². *Could not terminate.

Table 2: Results for SST dataset.

**Results for Videos** Table 1 gathers the results obtained for the three natural video datasets with artificial measurements (Raindrops, Remove-Pixel, and Moving-Vertical-Bar). For all measurements, the FID and FVD performance obtained by our model are 20%-50% better than DINEOF. This means that our model better controls the both spatial and the temporal generation quality than DINEOF. Globally, we achieve better MAE scores notably for color videos with few exceptions (performance

is close for Remove-Pixel). As for Newson et al. (2014), the completion could not be finished for Raindrops and Remove-Pixel masks for the same reason mentioned above caused by highly complex masks breaking cuboid patches. Newson et al. (2014) performs better when the form of masks is simple such as Moving-Vertical-Bar, for which matching patches could be easily found in neighbour frames. However, the computation time of Newson et al. (2014) is much longer than our model in terms of best performance could be achieved. Note that reduced computation time was an argument put forward in their publication. For a 30-frame $64\times64$ video, Newson et al. (2014) costs on average 1 minute and a GPU speedup is not available, versus around 0.5 seconds by our model, with an end-to-end GPU speedup.

**Comparison with Supervised Baselines**   Table 3 compares our model with the two supervised (unpaired and paired) variants described in Section 3.3. Unsurprisingly, the performance of supervised models is far better than the ones of our unsupervised model. We can find out that the access to the ground truth reduce dramatically all three metrics. By using supervision, FID is halved and FVD is between two

| Method | FID | FVD | MAE |
|---|---|---|---|
| Ours, Unsupervised | 43.72 | 1574.89 | .0834±.0187 |
| Unpaired, Supervised | **20.86** | **575.08** | **.0547±.0105** |
| Paired, Supervised | 22.17 | 720.75 | .0555±.0108 |

Table 3: Comparison with supervised baselines for FaceForensics++ with Raindrops.

and three times smaller. The error reduction is smaller with MAE. We also notice that the unpaired version performs better than the paired one in terms of sequence completion quality (FVD) as the $L^1$ loss introduces a strong constraint for the reconstruction. This shows to what extent the absence of ground truth will affect generation quality and the extra difficulty while dealing with partial observations.

## 4.2   Ablation Study

We also conduct additional experiments in order to quantify the importance of the temporal component.

In a first series of experiments, we remove the sequence component from our model, i.e. removing the sequence discriminator $D_s$ and replacing 3D generator by 2D one generating individually frames. Table 4 shows that our model clearly improves temporal quality by reducing FVD by a ratio of 7 compared to the

| Method | FID | FVD | MAE (°C) |
|---|---|---|---|
| Ours | **8.01** | **40.76** | **.0739±.0324** |
| Recurrent variant | 13.29 | 67.37 | .0960±.0431 |
| Static variant | 35.91 | 279.78 | .1036±.0047 |

Table 4: Comparison of results for SST data for ablation study.

model without the temporal component (denoted Static variant in the table). Note that FID is also clearly improved by a factor of 4. this gives more evidence that the model is able to exploit temporal dependency for its image completion task. We provide samples for this part in Appendix D in Figure 12. We further compared our model in Appendix E with Criminisi et al. (2004) and Ulyanov et al. (2017), two other state-of-the-art unsupervised image inpainting methods. The results show the same improvement on temporal quality as mentioned above.

Our model generates a frame at time $t$, $\hat{x}_t$ from a whole sequence of observations $y$. In a second series of experiments, we conditioned the generation of frames $\hat{x}_t$ only on past observations. We feed past observations into a convolutional RNN (we used GRU in our experiments) and generate the reconstructed frame, still denoted $G(y)$ by abuse of notation, from the last hidden state of the RNN, which encodes all past observations. The spatial discriminator operates as before, while the sequence discriminator operates on past observations only, instead of the full sequence of observations in our model. See Appendix C for an illustration and for further description. Results in Table 4 - Recurrent variant, show that using only past observations makes the completion less realistic and less accurate, but it still clearly outperforms the model without time dependency.

## 5   Related Work

There is currently, up to our knowledge, no other learning-based approach trying to solve the problem of spatiotemporal data completion in a purely unsupervised manner. We will review below related contributions for image and video reconstruction, data assimilation, and domain translation.

**Image Reconstruction**   Video or more generally spatiotemporal sequence completion can be considered as an extension of image completion problems. the first attempts for image completion and

inpainting were all supervised. Xie et al. (2012) uses convolutional NNs for regressing observations to ground truth images. This typically produces blurry outputs. To overcome this issue, some authors introduce textures (Yang et al., 2016), while many others make use of GANs (Pathak et al., 2016; Yu et al., 2018). More recently, unsupervised approaches have been developed by considering only corrupted images. Ulyanov et al. (2017); Lehtinen et al. (2018) show that it is possible to learn the underlying data distribution and to reconstruct images from observations when a model of observation process is given or when the noise is zero-mean. Such restrictive hypothesis have been removed in the seminal work of Bora et al. (2018). They introduce AmbientGAN to *unconditionally* generate images without supervision from corrupted observations under the assumption that the stochastic measurement process is known. MisGAN (Li et al., 2019) extend this idea and to learn jointly the mask and the original data distributions. Both contributions objective is data generation and not completion like we do here. Pajot et al. (2019) propose to *conditionally* recover images from corrupted observations only by solving a maximum *a posteriori* (MAP) estimation problem, implemented with an adversarial framework. This is limited to still images.

**Video Inpainting**  Video inpainting has been mainly considered in a supervised framework. Object-based (Cheung et al., 2006) and patch-based (Newson et al., 2014) approaches introduced before the deep learning era generally rely on prior segmentation of moving objects and background or strong assumptions on video content. Flow-based methods have been used to model spatial appearance and local pixel movement between consecutive frames. Huang et al. (2016) propose to guide non-parametric patch-based optimization with forward and backward optical flow. Xu et al. (2019); Kim et al. (2019) try to resolve the problem through neural optical flow estimation, which requires extra pre-trained network. More recently end-to-end learning approaches have been proposed. For example, Wang et al. (2018a) propose frame-level generation decomposition by combining a video inpainter with a frame-wise refinement inpainter. Extensions of image inpainting methods are also proposed in Chang et al. (2019). All these learning-based methods are trained with supervision and have been developed for natural videos.

**Data Assimilation for Remote Sensing**  For remote sensing applications, Optimal Interpolation (OI) is widely used in operational products (Donlon et al., 2012). It produces a linear estimate for the occluded area. Model-based assimilation methods (Ubelmann et al., 2015) rely on explicit physical dynamic priors and demand significant computational power. Purely data-driven methods based on empirical orthogonal functions (EOF, Beckers & Rixen, 2003) use basically matrix factorization to achieve temporal interpolation. Recent advances in Analog Data Assimilation (AnDA, Lguensat et al., 2017; Fablet et al., 2018) combine analog forecasting methods with data-driven assimilation using implicit knowledge of dynamical prior. These methods rely either on interpolation or exploit some priors on the nature of the underlying process. Recently, learning methods have started to be exploited in this field. Shibata et al. (2017) propose to apply learning-based frame-level inpainting enhanced with optical flow using simple assumptions on pixel movement. In a later paper, Shibata et al. (2018), they recover the missing data using an adversarial approach to supervise extra occluded area w.r.t the original partial observations. This approach still reconstruct data frame by frame.

**Domain Translation**  Reconstruction can also be considered as a translation problem between two domains, incomplete observations and full unobserved data. For images, Pix2Pix (Isola et al., 2016) utilizes GANs to project data from domain A to domain B with paired data. CycleGAN (Zhu et al., 2017) propose to use two generator-discriminator pairs to model the transformation between two domains. For videos, Wang et al. (2018b) propose Vid2Vid by adding a multi-scale temporal discriminator in Pix2Pix to supervise the optical flow. RecycleGAN (Bansal et al., 2018) is based on the idea of CycleGAN by adding a temporal transformation in both domains. However, these methods require full data from the two domains, and sometimes the supervision on motion, when no supervision is available in our setting.

## 6  CONCLUSION

We have proposed a GAN-based framework to complete partially observed spatiotemporal data. Our model utilizes a generator to complete missing pixels in observation sequences with the help of two discriminators classifying real and generated observation sequences. We show that our model is able to complete spatiotemporal data without ground truth supervision when we have a stochastic

model of the occlusion process. Our results for SST data and natural videos show that the recovered sequences are realistic, especially when the occluded area is highly complex.

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

## A    SUPPLEMENTARY INFORMATION ON SST DATASET

GLOBAL Sea Physical Analysis and Forecasting Product is a monitor system providing simulated but realistic global ocean SST data. In analysis engine, it integrates satellite-derived and *in situ* data by assimilation from OSTIA SST system (OSTIA, The Operational Sea Surface Temperature and Sea Ice Analysis, Donlon et al., 2012). The analysis is based on a sophisticated ocean model, NEMO ocean engine (Nucleus for European Modeling of the Ocean, Gurvan et al., 2017), a state-of-the-art modelling framework of ocean related physics. Oceanic output variables of this product are hourly, daily and monthly means of temperature, salinity, currents, sea surface height, etc.

Our dataset is a subset of `global-analysis-forecast-phy-001-024-hourly-t-u-v-ssh`, which includes hourly mean Sea Temperature at sea level height, i.e. SST. The data we use is a part of the archive of analysis integrating real-world data. Training-and-validation set is hourly SST in 2018 (2018-01-01 00:30 to 2018-12-31 23:30) on a marine region of $64 \times 64$ pixels (20°-25.25°N, 34.75°-40°W, in North Atlantic Ocean). Test set corresponds to the data of the first 60 days of 2019 (2019-01-01 00:30 to 2019-03-01 23:30) in another region of the same size (14.75°-20°S, 14.75°-20°W, in South Atlantic Ocean). To obtain the dataset, check out the product via `http://marine.copernicus.eu/services-portfolio/access-to-products/?option=com_csw&view=details&product_id=GLOBAL_ANALYSIS_FORECAST_PHY_001_024`, then enter the criteria above for each subset.

## B    ARCHITECTURE DETAILS

We detail the architecture of networks utilized in our experiments in this section.

**Generator**    Table 5a lists modules for generator $G$. It is a ResNet-type self-attention network as in Zhang et al. (2019). It is composed of 3D ResNet blocks as in He et al. (2016) and spatial self-attention layer from Wang et al. (2018c), which means the attention calculation is limited within each frame.

**Discriminators**    Table 5b is the architecture of 2D or 3D PatchGAN discriminator as in Isola et al. (2016). Frame discriminator $D_f$ is a 2D convolutional NN. Sequence discriminator $D_s$ uses the same structure as $D_f$ but with 3D convolutions except that the stride is still 1 for temporal dimension, such that the number of frames is not limited.

## C    DETAILS OF VARIANTS OF OUR MODEL

We describe two supervised variants as supervised baselines in Section 4.1 and a recurrent variant of our model in Section 4.2.

**Unpaired Variant, Supervised (Figure 3a)**    This variant gives our model access to the distribution $p_{\mathbf{X}}$. Instead of distinguishing between true observations $\boldsymbol{y}$ and fake ones $\hat{\boldsymbol{y}}$, $D_f$ and $D_s$ will discriminate between original sequences $\boldsymbol{x}$ and the output of the generator $\hat{\boldsymbol{x}}$.

|  | Module | Nb. Input Channel | Nb. Output Channel | Activation |
|---|---|---|---|---|
|  | Encoder | | | |
| 1 | 3D ResNet block | $C_{img}$ | $C_{base}$ | ReLU* |
| 2 | 3D ResNet block | $C_{base}$ | $16C_{base}$ | ReLU* |
| 3 | 3D ResNet block | $16C_{base}$ | $16C_{base}$ | ReLU* |
|  | Decoder | | | |
| 4 | 3D ResNet block | $16C_{base}$ | $8C_{base}$ | ReLU* |
| 5 | 3D ResNet block | $8C_{base}$ | $4C_{base}$ | ReLU* |
| 6 | 3D ResNet block | $4C_{base}$ | $2C_{base}$ | ReLU* |
| 7 | Spatial Self-Attention | $2C_{base}$ | $2C_{base}$ | ReLU* |
| 8 | 3D ResNet block | $2C_{base}$ | $C_{base}$ | ReLU* |
| 9 | 3D Batch Norm. | $2C_{base}$ | $2C_{base}$ | ReLU |
| 10 | 3D Conv. | $C_{base}$ | $C_{img}$ | tanh |

(a) Generator structure. Kernel size 3, stride 1. *Activation inside the module

|  | Module | Nb. Input Channel | Nb. Output Channel | Spatial Stride | Activation |
|---|---|---|---|---|---|
| 1 | 2D/3D Conv. | $C_{img}$ | $C_{base}$ | 2 | LeakyReLU[†] |
| 2 | 2D/3D Conv. | $C_{base}$ | $2C_{base}$ | 2 | LeakyReLU[†] |
| 3 | 2D/3D Conv. | $2C_{base}$ | $4C_{base}$ | 2 | LeakyReLU[†] |
| 4 | 2D/3D Conv. | $4C_{base}$ | $8C_{base}$ | 2 | LeakyReLU[†] |
| 5 | 2D/3D Conv. | $8C_{base}$ | $8C_{base}$ | 2 | LeakyReLU[†] |
| 6 | 2D/3D Conv. | $8C_{base}$ | $8C_{base}$ | 2 | LeakyReLU[†] |
| 7 | 2D/3D Conv. | $8C_{base}$ | $8C_{base}$ | 1 | LeakyReLU[†] |
| 8 | 2D/3D Conv. | $8C_{base}$ | $8C_{base}$ | 1 | — |

(b) PatchGAN Discriminator. Kernel size 3. Stride of temporal dimension is always 1. 2D convolution for frame discriminator $D_f$, 3D for sequence one $D_s$. [†]Negative slope 0.2.

Table 5: Architecture of networks.

**Paired Variant, Supervised (Figure 3b)**   This variant gives our model access not only to the distribution $p_\mathbf{X}$ but also the joint distribution $p_{\mathbf{X},\mathbf{Y}}$ by adding a sequence-to-sequence $L^1$ reconstruction loss. We use the duo-discriminator setting as our model to prevent generating blurry frames, which is essentially the same techniques in Isola et al. (2016) and Wang et al. (2018b).

**Recurrent Variant, Unsupervised (Figure 4)**   In this variant, we simply insert a convolutional recurrent network cell of any type, ConvGRU in our case (Convolutional Gated Recurrent Unit, Siam et al., 2016), into Pajot et al. (2019). See Appendix C for the illustration and further description. This cell models temporal dependency in feature maps after encoding the sequence into the space with maximum number of channels. Instead of discriminating image by image as in Pajot et al. (2019), we use in this variant the same duo-discriminator setting as our model. $D_f$ distinguishes frame by frame while $D_s$ discriminates true sequence clip $\boldsymbol{y}_{t-L+1}^t$ and fake clip $\hat{\boldsymbol{y}}_{t-L+1}^t$, where $L$ is the maximum number of consecutive frames.

# D   ADDITIONAL SAMPLES

We provide more longer samples from with samples from baselines in this section, see figures 5, 6, 7, and 8. We also show extra samples for unsupervised baselines (Figure 10), supervised baselines (Figure 11), ablation study (Figure 12).

# E   ADDITIONAL COMPARISON WITH STATE-OF-THE-ART UNSUPERVISED IMAGE INPAINTING APPROACHES

We provide here the additional comparison of results with two state-of-the-art unsupervised inpainting methods, Criminisi et al. (2004) and Ulyanov et al. (2017), for FaceForensics++ dataset with Raindrops noise. Unsurprisingly, Table 6 shows that our model outperforms these methods, especially in terms of temporal quality indicated by FVD. We show also some samples in Figure 13.

| Method | FID | FVD | MAE |
|---|---|---|---|
| Criminisi et al. (2004) | 147.86 | 3617.92 | .5533±.1246 |
| Ulyanov et al. (2017) | 44.84 | 2410.62 | .2271±.1560 |
| **Ours** | **43.72** | **1574.89** | **.0834±.0187** |

Table 6: Additional comparison with state-of-the-art unsupervised image inpainting methods for FaceForensics++ with Raindrops.

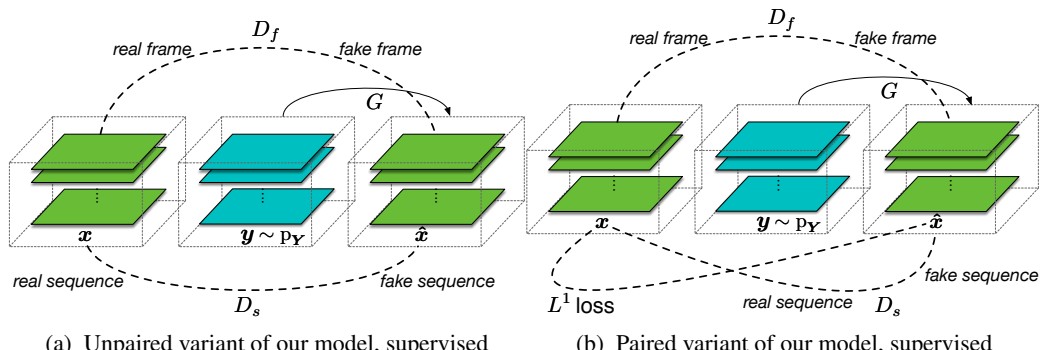

(a) Unpaired variant of our model, supervised  (b) Paired variant of our model, supervised

Figure 3: Supervised variants, used as baseline in Section 3.3

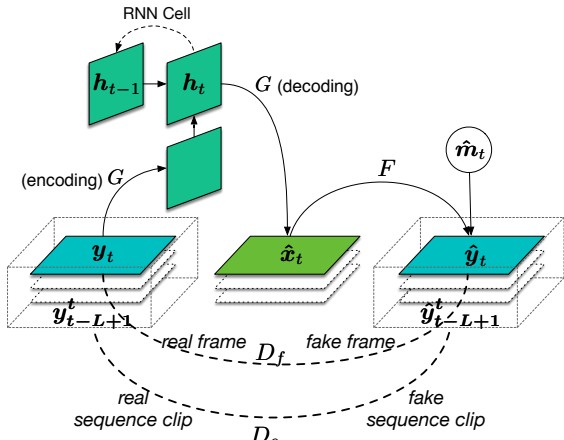

Figure 4: Recurrent variant.

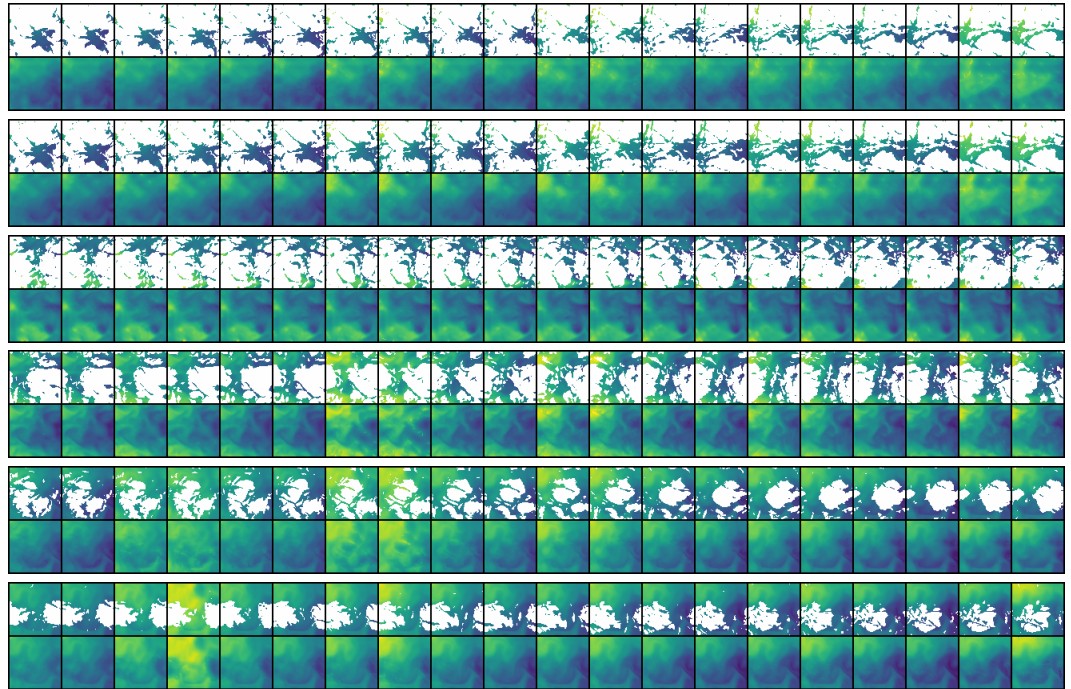

Figure 5: Samples from our model for SST. From top to bottom: Cloud at LWP threshold 55, 60, 65, 70, 75, and 80 g/m$^2$.

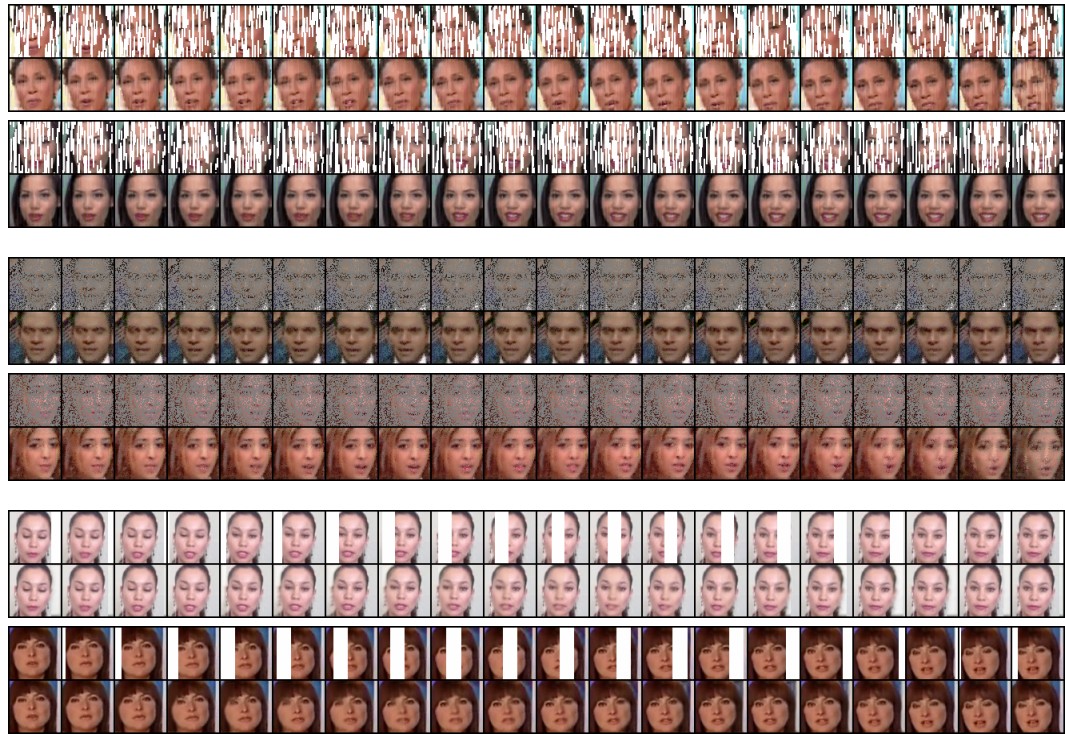

Figure 6: Samples from our model for FaceForensics++. From top to bottom: Raindrops, Remove-Pixel, and Moving-Vertical-Bar.

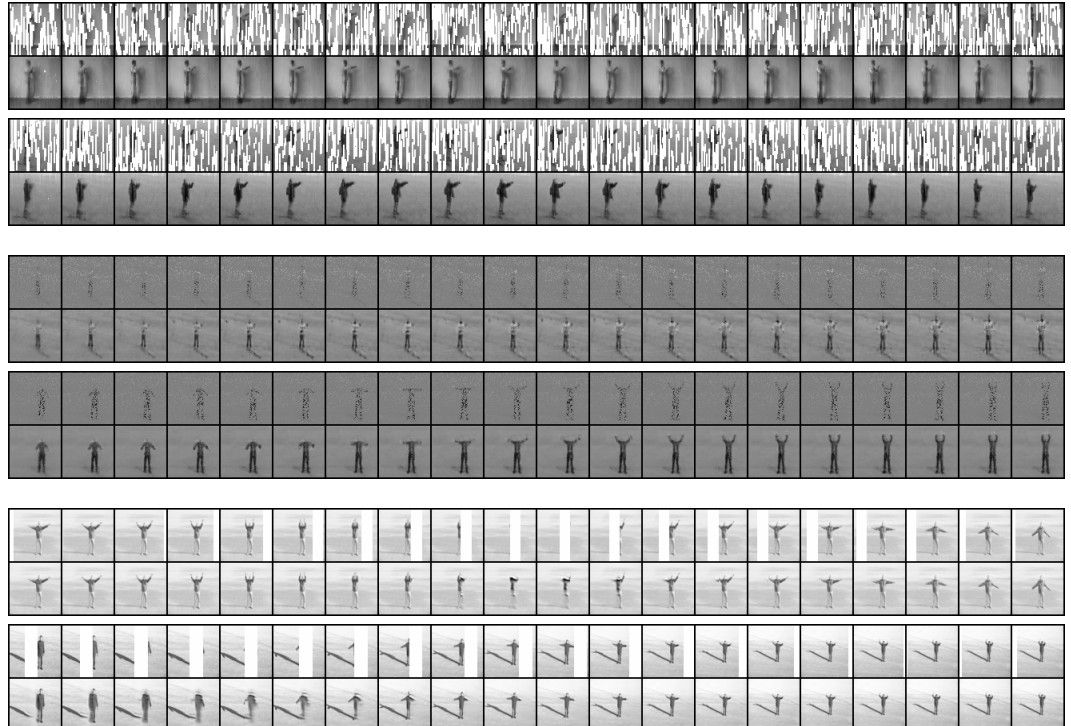

Figure 7: Samples from our model for KTH. From top to bottom: Raindrops, Remove-Pixel, and Moving-Vertical-Bar.

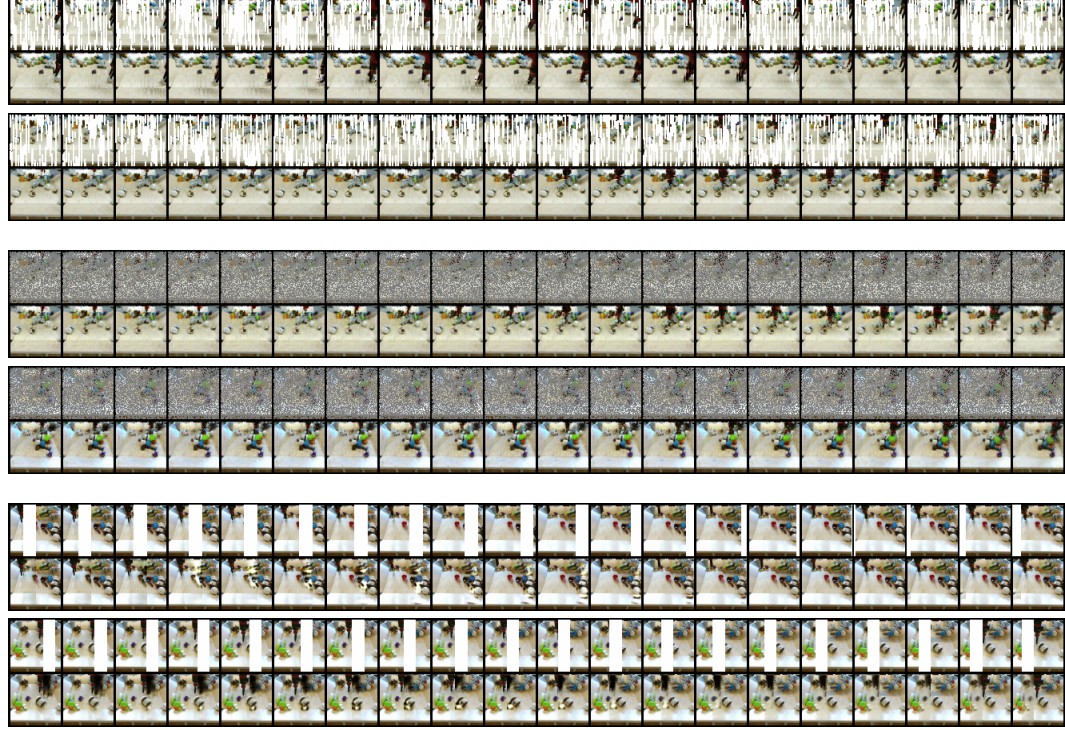

Figure 8: Samples from our model for BAIR. From top to bottom: Raindrops, Remove-Pixel, and Moving-Vertical-Bar.

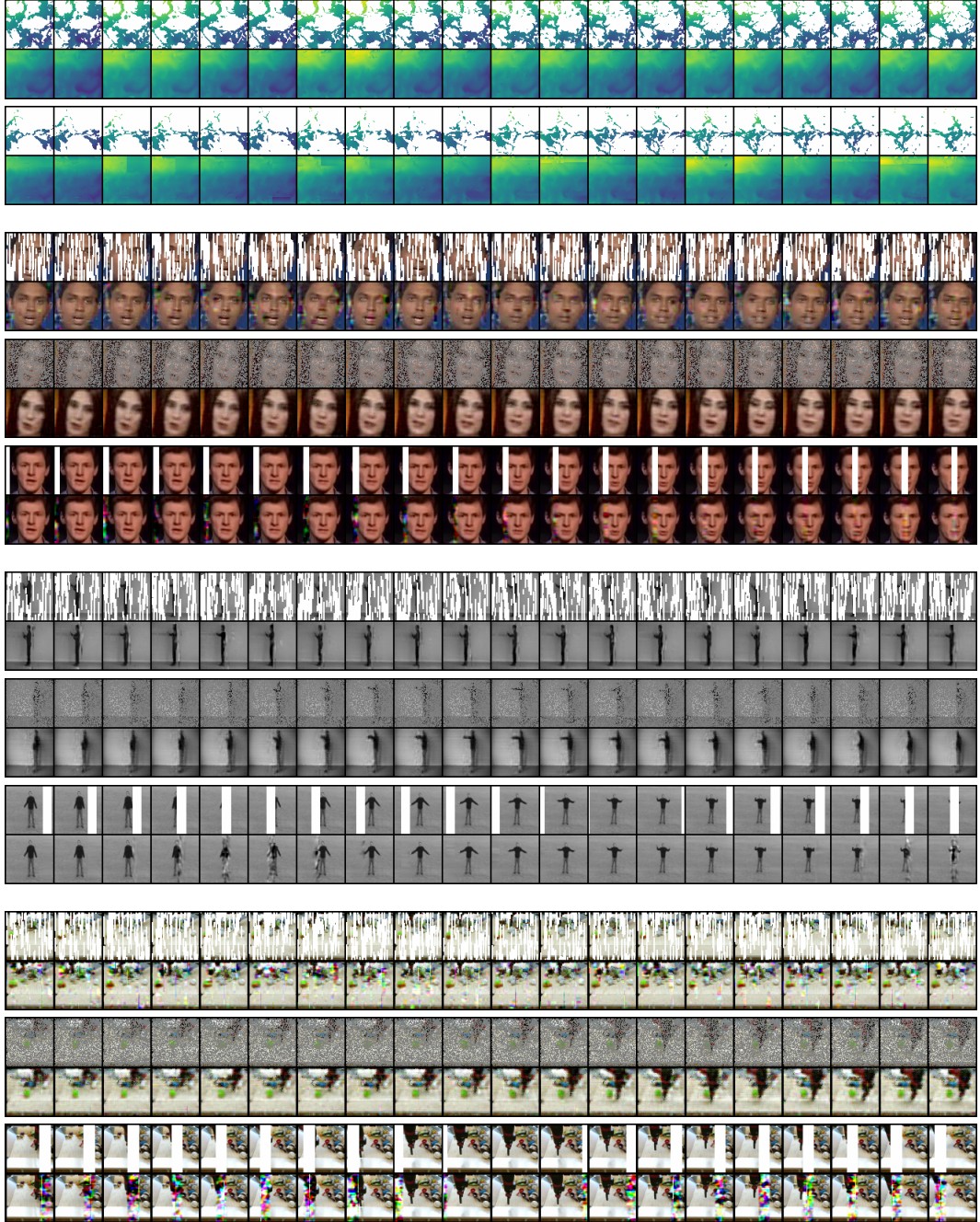

Figure 9: Samples from DINEOF (Alvera-Azcárate, 2011) for SST, FaceForensics++, KTH and BAIR.

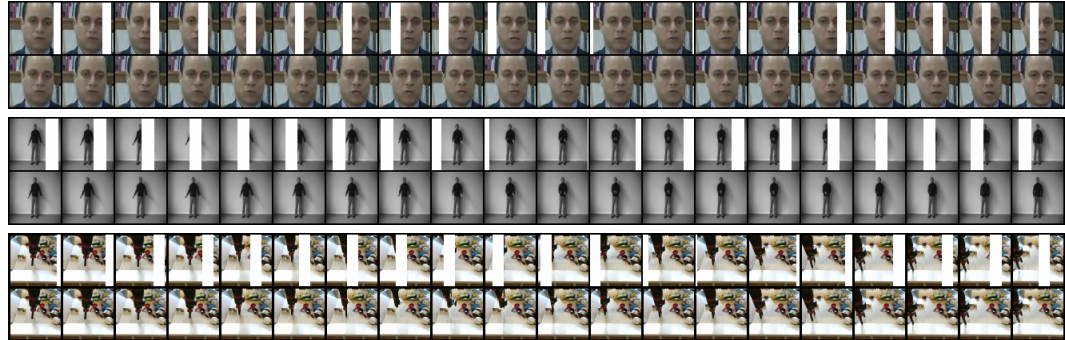

Figure 10: Samples from Newson et al. (2014) for FaceForensics++, KTH, and BAIR. Only with Moving-Vertical-Bar. Samples for other measurements cannot be calculated in reasonable time.

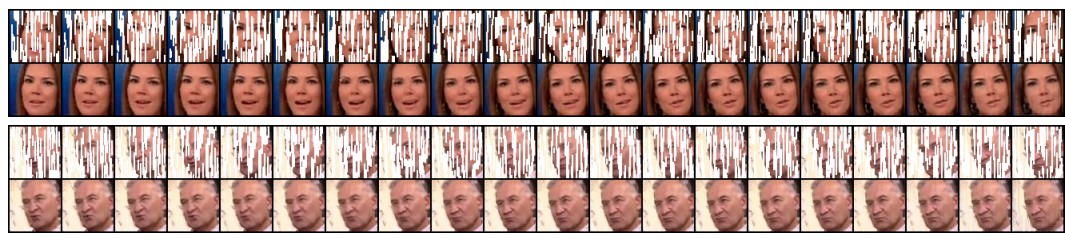

(a) Unpaired variant

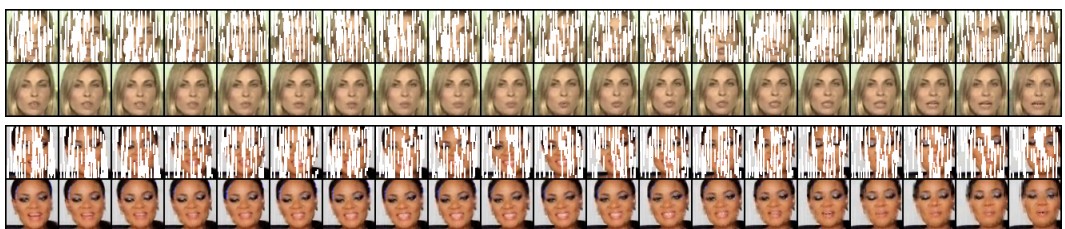

(b) Paired variant

Figure 11: Samples from supervised variants.

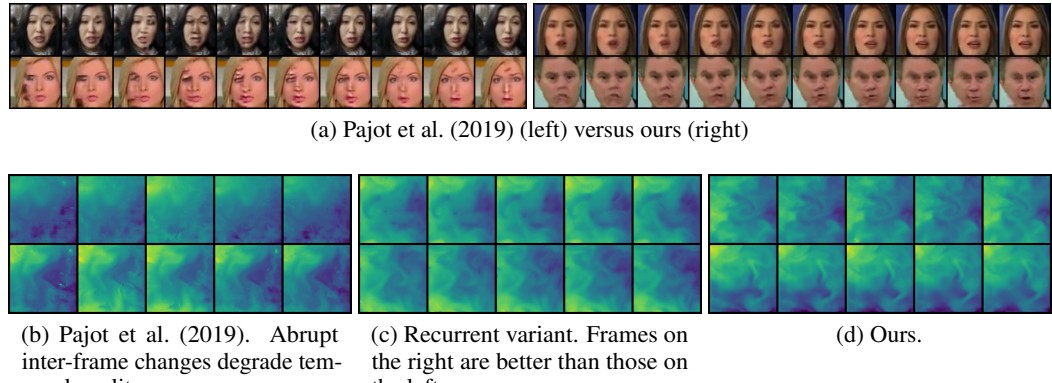

(a) Pajot et al. (2019) (left) versus ours (right)

(b) Pajot et al. (2019). Abrupt inter-frame changes degrade temporal quality.

(c) Recurrent variant. Frames on the right are better than those on the left.

(d) Ours.

Figure 12: Samples for ablation study.

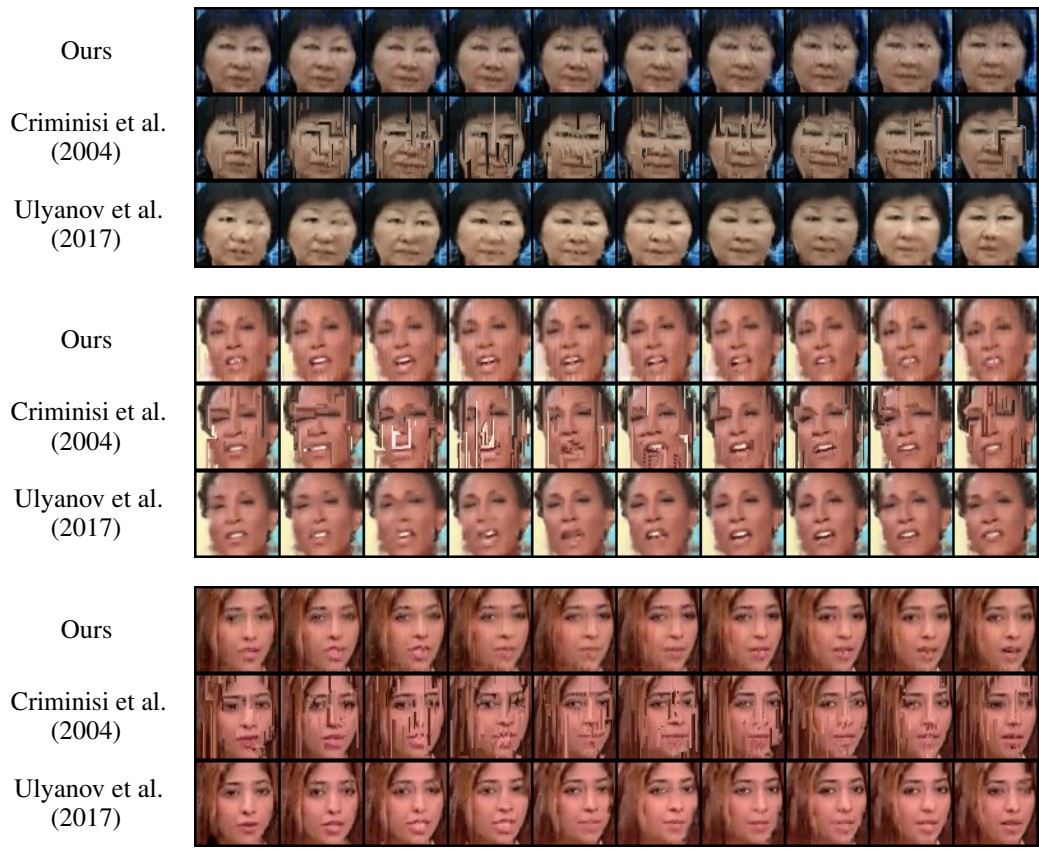

Figure 13: Comparison of samples from our model and two state-of-the-art unsupervised image inpainting methods. Note that the temporal coherence is broken for Ulyanov et al. (2017) due to the instant appearance and disappearance of facial parts.

