# OpenReview forum: "Unsupervised Spatiotemporal Data Inpainting"
_ICLR.cc/2020/Conference — Reject_

### Official Review · AnonReviewer2 · 2019-10-17
**Official Blind Review #2**

**Rating:** 3

**Review:**

This paper proposes a GAN based approach for unsupervised video inpainting. The idea is to learn an auto encoder to reconstruct the unoccluded regions of the frames, and reuse the autoencoder as the generator for a video GAN, which discriminates between real noisy videos and generated videos masked by a known mask distribution.

To me the general idea is a straightforward application of GANs to the general inpainting/denoising problem, with the novelty being 1) not relying on paired or unpaired references and 2) extension to the video domain. The main critique I have is that the work seems incomplete and does not show great empirical results, which makes this work a much weaker contribution. Obviously, the authors should complete the experimental results on baseline method 2. It'd also be informative to compare with SOTA unsupervised image inpainting algorithms, if more such baselines are available than video based ones.

Another limitation of this work is that a known mask distribution is assumed, while this might be a reasonable assumption in some cases, it can also be easily violated in real world problems. It'd be great improvement if such an assumption can be relaxed, for example, assuming a family of mask distributions but not the exact instantiation.

Overall I think this work attempts to solve an interesting problem with an incremental but reasonable approach, and more empirical evaluation is needed to make the paper meet the bar for acceptance.

**Experience Assessment:**

I have published one or two papers in this area.

**Review Assessment: Checking Correctness Of Derivations And Theory:**

I assessed the sensibility of the derivations and theory.

**Review Assessment: Checking Correctness Of Experiments:**

I assessed the sensibility of the experiments.

**Review Assessment: Thoroughness In Paper Reading:**

I read the paper at least twice and used my best judgement in assessing the paper.

---

> ### Author Response · Authors · 2019-11-15
> **Reply to Blind Review #2**
>
> Thank you for your kind and helpful comments. We provide below some clarifications.
>
> - Comments on the Experiments
>
> We tried hard to perform the experiments for baseline (2) [1]. We here explain in detail its results shown in tables 1 and 2.
>
> [1] searches for the nearest neighbours of an occluded area using an Approximate Nearest Neighbor (ANN) search with the PatchMatch algorithm [11]. The occluded area is then reconstructed by assembling information from these neighbors at multiple scales. To initialize the algorithm, PatchMatch needs to find a valid initial guess for each occluded pixel, which indicates where to find the corresponding patch in the sequence. This initialization will not terminate until every occluded pixel is pointed to a patch who does not include any other occluded pixel.
>
> Since [1] specifically looks for rectangular cuboids of video information, it is extremely well adapted for the Moving-Vertical-Bar and thus performs well. However, for more general complex types of noise such as Raindrops, Remove-Pixel, and Cloud noises, it cannot work properly. More precisely, it remains blocked in the search for relevant candidate patches for the occluded pixels. This means that no matter how hard it tries, it cannot complete the task. The absence of the experiment results for [1] in tables 1 and 2 is due to this issue and not to insufficient running time. For example, we also tried with very small patches (3x3x3), a reasonable minimum size for spatiotemporal patches, the algorithm still remains blocked and is then unable to terminate. This was probably not clear enough in the original manuscript and this will be made explicit in a new version.
>
> - Additional Experimental Results
>
> As suggested,we performed additional tests using [2] and [3], two SOTA unsupervised image inpainting approaches. The results are provided in the table below and have been added in appendix E in the paper. The table shows quantitative results for  [2, 3] and our approach for the FaceForensics++ dataset with Raindrops noise among, and our model. We show that the inpainting results of [2, 3] are quantitatively worse, especially for temporal quality metric FVD.
>
> -------------------------------------------------------------
>           |    FID     |     FVD      |          MAE
> -------------------------------------------------------------
>    [2]   |   44.84  |  2410.62   |  0.2271±0.1560
>    [3]   | 147.86  |  3617.92   |  0.5533±0.1246
> -------------------------------------------------------------
> Ours  |   43.72   |  1574.89  |  0.0834±0.0187
> -------------------------------------------------------------
>
> A qualitative illustration of the behavior of these models has been added to the following site, they confirm their poor performance: https://sites.google.com/view/unsup-video-inpaiting/.
>
> - On the Mask Distribution Hypothesis
>
> The assumption of a specific observation model (i.e. in our case, the known mask distribution)  for image imputation/denoising is widely adopted in the vision and deep learning community  [2, 7, 8] and also in the physical modeling community  [4-6]. For  unsupervised video inpainting approaches, even stronger assumptions on the content and/or the form of masked and unmasked region are often used. For example the assumption of the existence of a object-background segmentation in [9] or the existence of spatiotemporal patches in [1]. Our assumptions follow this line of research. Recently,  MisGAN [10] relaxed this assumption, in the case when missing pixel position can be identified. This could be an extension of our work.
>
>
> References:
> [1] Alasdair Newson, Andrés Almansa, Matthieu Fradet, Yann Gousseau, and Patrick Pérez. Video inpainting of complex scenes.
> [2] Dmitry Ulyanov, Andrea Vedaldi, and Victor S. Lempitsky. Deep Image Prior.
> [3] Antonio Criminisi, Patrick Pérez, and Kentaro Toyama. Region filling and object removal by exemplar-based image inpainting.
> [4] Unified Notation for Data Assimilation. Operational, Sequential and Variational
> [5] M. Bocquet, C.A. Pires, and L. Wu. Beyond Gaussian Statistical Modeling in Geophysical Data Assimilation.
> [6] R. Lguensat, P. Tandeo, P. Ailliot, M. Pulido, and R. Fablet. The Analog Data Assimilation.
> [7] Ashish Bora, Eric Price, and Alexandros G. Dimakis. AmbientGAN: Generative models from lossy measurements.
> [8] Jaakko Lehtinen, Jacob Munkberg, Jon Hasselgren, Samuli Laine, Tero Karras, Miika Aittala, and Timo Aila. Noise2Noise: Learning image restoration without clean data.
> [9] S. S. Cheung, J. Zhao, and M. V. Venkatesh. Efficient object-based video inpainting.
> [10] Steven Cheng-Xian Li, Bo Jiang, Benjamin M. Marlin. MisGAN: Learning from Incomplete Data with Generative Adversarial Networks.
> [11] Connelly Barnes, Eli Shechtman, Adam Finkelstein, Dan B. Goldman. PatchMatch: A Randomized Correspondence Algorithm for Structural Image Editing

---

### Official Review · AnonReviewer1 · 2019-10-20
**Official Blind Review #1**

**Rating:** 3

**Review:**

Summary
The paper presents an approach to perform video inpainting from corrupted input only . Their approach proposed a method that uses GANs for denoising images to now handle full sequences of images by building on top work that handles inpainting in single images.

Strengths
1) The authors present extensive experiments on many datasets.
2) The presented approach is simple and general enough to be applied to many video problems.
3) The provided code is well structured and easy to read and atrached webpage showcases their results well.
4) The paper is well-written.

Weakness
1) From the code and the description in paper, it seems that a different corruption (https://github.com/anon-ustdi/ustdi/blob/7a81db4972ef9d4eabbd8fe354a8984a7771ae5d/src/datasets/corrupted.py) is applied for each step. Can the authors confirm if that is the case?

If this is the case, I am afraid the method cannot be called unsupervised as across many steps the model would have seen different corruptions of the same video and across many such corruptions the model can learn what an uncorrupted video looks like. It is okay if that is the case but the claim of unsupervised would not hold. If that is the case, the authors need to make comparison with other supervised inpainting methods as well [1,2]

2) Is there a dataset for which these corruptions exist naturally? May be [3, 4].  It would be nice to have experiments on a dataset where the corruptions are present naturally.

3) While the experiments are incomplete for Table 1,  [4] outperforms the proposed approach for the one dataset the numbers have been reported. Performance comparison with [4] are not fair as [4]'s implementation is in MATLAB.

Decision
While the presented approach is good, further experiments are required to further validate the effectiveness of their approach in an unsupervised setting.

References
[1] Chuan Wang, Haibin Huang, Xiaoguang Han, and Jue Wang. Video inpainting by jointly learning temporal structure and spatial details.
[2] Dahun Kim, Sanghyun Woo, Joon-Young Lee, and In So Kweon. Deep video inpainting.
[3] https://github.com/stayhungry1/Video-Rain-Removal
[4] https://github.com/nnUyi/DerainZoo
[5] Alasdair Newson, Andrés Almansa, Matthieu Fradet, Yann Gousseau, and Patrick Pérez. Video inpainting of complex scenes.

**Experience Assessment:**

I have read many papers in this area.

**Review Assessment: Checking Correctness Of Derivations And Theory:**

N/A

**Review Assessment: Checking Correctness Of Experiments:**

I carefully checked the experiments.

**Review Assessment: Thoroughness In Paper Reading:**

I read the paper at least twice and used my best judgement in assessing the paper.

---

> ### Author Response · Authors · 2019-11-15
> **Reply to Blind Review #1 (Part 1)**
>
> Thank you for your kind and helpful remarks. We provide below clarifications for your questions.
>
> 1) You are right, a different mask is applied at different time steps. This mask could be time independent (as in Remove-Pixel noise) or time-dependent (for the three other noises: Raindrops, Moving-Vertical-Bar and Cloud noises). The underlying hypothesis is that in many practical situations, the noise is a dynamic and time-evolving process. Alternatively, we could have considered a fixed noise pattern per sequence, in which case our formalism is still valid, or a fixed noise pattern for all the sequences, in which case it is not applicable anymore (because of the non stochasticity of the noise position process). Note that our hypothesis is similar to the one used in the classical data imputation setting in statistics [6, 7], for non dynamic data.  This is also the usual assumption in the GAN literature (again on still data): for example in [8, 9] the missing data process is supposed to be stochastic in the sense that the position of the missing data follows a given distribution. The model will then have a chance of observing the whole data distribution, provided it has access to enough data.
>
> The problem is still considered unsupervised because there is no direct supervision, be it paired (noisy image, ground truth image) or unpaired noisy sequences and ground truth sequences without direct correspondence between the two types of sequences. More details are provided below.
>
> (a) Unsupervised versus supervised:
>
> Your remark is perfectly relevant since given enough data, one can imagine that the model will have access to the whole information from the videos. However, i) this would require an extremely large number of observed sequences, ii) this is an extremely complex task when no guidance or prior is provided to the model. For unsupervised inpainting, the model has to discover by itself which information is relevant for the reconstruction (in this case local spatiotemporal information).
>
> In [1, 2], for example, the inpainter is trained with ground truth information. More than that, the inpainter has access to additional information about pixel displacement through optical flow estimation performed by FlowNets [10, 11]. This further encourages the model to make use of neighbor image information for the reconstruction. Note that optical flow is not available in our case because of the nature of the noise itself.
>
> Alternative frameworks based on GANs could be used such as Pix2Pix (Vid2Vid for videos) or variants of CycleGAN (RecycleGAN for videos). These methods have been developed for image or video translation and could be used as well for imputation. However, the former relies on paired supervision and the latter on unpaired supervision, which again are not available in our case.
>
> Note that for time-dependent slowly moving masks like Clouds at heavy coverage, there exist numerous areas that will never been seen in the sequence, and our model can still recover the masked area with good spatial and temporal quality.
>
> (b) Comparison with supervised baselines:
>
> We propose in the paper a comparison with two supervised variants of our model. One makes use of paired supervision and the other one of unpaired supervision. This is addressed in Section 3.3 for a description of the model variants and in Section 4.1 (last paragraph) for a quantitative comparison. Supervision brings a lot of additional information and considerably improves the model performance. However, in most cases such a supervision is never available and this analysis has been performed only to show the performance gap w.r.t an ideal situation.
>
> 2) Thank you for proposing a large number of existing de-rain datasets. However, even if we consider the datasets for frame-by-frame de-raining, the associated mask distributions in [3, 4] are not natural. They are often simulated with control parameters, as for instance in [3, 12]. Some of the methods have a natural subset of rainy images, but no annotated mask comes with the datasets. They are usually used uniquely while testing after training the model with synthetic masks, such as in [13, 14]. Note that our Cloud measurement is already a quite realistic mask distribution based on very sophisticated large-scale cloud simulation, showing that our model is capable to work in the real world application.

---

> ### Author Response · Authors · 2019-11-15
> **Reply to Blind Review #1 (Part 2)**
>
> 3) Let us first introduce some more details on [5].
>
> [5] searches for the nearest neighbours of an occluded area using an Approximate Nearest Neighbor (ANN) search with the PatchMatch algorithm [15]. The occluded area is then reconstructed by assembling information from these neighbors at multiple scales. To initialize the algorithm, PatchMatch needs to find a valid initial guess for each occluded pixel, which indicates where to find the corresponding patch in the sequence. This initialization will not terminate until every occluded pixel is pointed to a patch which does not include any other occluded pixel.
>
> Since [5] specifically looks for rectangular cuboids of video information, it is extremely well adapted for the Moving-Vertical-Bar and thus performs well. However, for more general complex types of noise such as Raindrops, Remove-Pixel, and Cloud noises, it cannot work properly. More precisely, it remains blocked in the search for relevant candidate patches for the occluded pixels. This means that no matter how hard it tries, it cannot complete the task. The absence of the experiment results for [5] in tables 1 and 2 is due to this issue and not to insufficient running time. For example, we also tried with very small patches (3x3x3), a reasonable minimum size for spatiotemporal patches, the algorithm still remains blocked and is then unable to terminate. This was probably not clear enough in the original manuscript and this will be made explicit in a new version.
>
> For the concern on the performance comparison, even though the official code of [5] is written in MATLAB, it uses essentially C++ to achieve optimal performance on CPU. However, due to the nature of [5]’s iterative algorithm, an end-to-end acceleration by GPU is not possible. For each inference, it must search in the whole sequence for a small patch for each missing point, which naturally requires a lot of computing. As it is also iterative, the algorithm cannot be parallelized between different iterations and different substeps of each iteration. In contrast, as our inpainter is a neural network, it can fully benefit from GPU speedup leading the inference time down to around 0.5 seconds per sequence. We will update our manuscript to better explain this comparison.
>
>
> References:
> [5] Alasdair Newson, Andrés Almansa, Matthieu Fradet, Yann Gousseau, and Patrick Pérez. Video inpainting of complex scenes.
> [6] Zoubin Ghahramani, and Michael I. Jordan. Supervised learning from incomplete data via an EM approach.
> [7] Roderick J. A. Little, and Donald B. Rubin. Statistical Analysis with Missing Data.
> [8] Jinsung Yoon, James Jordon, Mihaela van der Schaar. GAIN: Missing Data Imputation using Generative Adversarial Nets.
> [9] Ashish Bora, Eric Price, and Alexandros G. Dimakis. AmbientGAN: Generative models from lossy measurements.
> [10] Philipp Fischer, Alexey Dosovitskiy, Eddy Ilg, Philip Häusser, Caner Hazirbas, Vladimir Golkov, Patrick van der Smagt, Daniel Cremers, Thomas Brox. FlowNet: Learning Optical Flow with Convolutional Networks.
> [11] Eddy Ilg, Nikolaus Mayer, Tonmoy Saikia, Margret Keuper, Alexey Dosovitskiy, Thomas Brox: FlowNet 2.0. Evolution of Optical Flow Estimation with Deep Networks.
> [12] Ruoteng Li, Loong Fah Cheong, Robby T. Tan. Heavy Rain Image Restoration: Integrating Physics Model and Conditional Adversarial Learning.
> [13] He Zhang, Vishwanath Sindagi, Vishal M. Patel. Image De-raining Using a Conditional Generative Adversarial Network.
> [14] Wenhan Yang, Robby T. Tan, Jiashi Feng, Jiaying Liu, Zongming Guo, Shuicheng Yan. Deep Joint Rain Detection and Removal from a Single Image
> [15] Connelly Barnes, Eli Shechtman, Adam Finkelstein, Dan B. Goldman. PatchMatch: A Randomized Correspondence Algorithm for Structural Image Editing

---

### Official Review · AnonReviewer3 · 2019-10-27
**Official Blind Review #3**

**Rating:** 6

**Review:**

The paper addresses the problem of reconstructing a video sequence that contains occlusions. The paper focusses on remove four particular types of simulated occlusion - Raindrops, Remove  pixel, Cloud and Vertically moving Bar. The authors show good results in reconstructing these video sequences in an unsupervised manner. The paper uses a GAN based network to accomplish inpainting in the occluded regions.  The authors claim that the method is very flexible in terms of the data that needs to come in, and test this by deploying the method to solve quite different missing data problems (Using different type of occlusion and in different contexts).

One interesting note about the experiments is that the presented method is outperformed by Newson et. al[2014] in all experiments where Newson's method does not complete. I think we need more data for this to be a reasonable constraint. How long was the experiment allowed to run without completing? Do you think there is a reasonable prospect that it could finish, given more time? Other than that the method performs extremely well across all tasks.

The paper is well written, and the methods and experiments are convincing. Another very good aspect of this paper is that the accompanying website is very good and contains code for the methods and experiments used.  The work is also sufficiently novel, and has some clever tricks (such as having the discriminator consider the inter frame difference).

Minor issues
On  pg 7 - I am unfamiliar with the uses of sota, and my intuition is that is should perhaps be SOTA. If this is common usage, please ignore me :)
On pg 9 - framwork (typo)


**Experience Assessment:**

I do not know much about this area.

**Review Assessment: Checking Correctness Of Derivations And Theory:**

I assessed the sensibility of the derivations and theory.

**Review Assessment: Checking Correctness Of Experiments:**

I assessed the sensibility of the experiments.

**Review Assessment: Thoroughness In Paper Reading:**

I read the paper at least twice and used my best judgement in assessing the paper.

---

> ### Author Response · Authors · 2019-11-15
> **Reply to Blind Review #3**
>
> Thank you for your comments and questions. Below we address your remarks.
>
> Newson et al. (2014) [1] is one of the SOTA patch-based methods for unsupervised inpainting. It searches for the nearest neighbours of an occluded area using an Approximate Nearest Neighbor (ANN) search with the PatchMatch algorithm [2]. The occluded area is then reconstructed by assembling information from these neighbors at multiple scales. To initialize the algorithm, PatchMatch needs to find a valid initial guess for each occluded pixel, which indicates where to find the corresponding patch in the sequence. This initialization will not terminate until every occluded pixel is pointed to a patch who does not include any other occluded pixel.
>
> Since [1] specifically looks for rectangular cuboids of video information, it is extremely well adapted for the Moving-Vertical-Bar and thus performs well. However, for more general complex types of noise such as Raindrops, Remove-Pixel, and Cloud noises, it cannot work properly. More precisely, it remains blocked in the search for relevant candidate patches for the occluded pixels. This means that no matter how hard it tries, it cannot complete the task. The absence of the experiment results for [1] in tables 1 and 2 is due to this issue and not to insufficient running time. For example, we also tried with very small patches (3x3x3), a reasonable minimum size for spatiotemporal patches, the algorithm still remains blocked and is then unable to terminate. This was probably not clear enough in the original manuscript and this will be made explicit in a new version.
>
> Minor comments: sure this is SOTA, thanks.
>
> References:
> [1] Alasdair Newson, Andrés Almansa, Matthieu Fradet, Yann Gousseau, and Patrick Pérez. Video inpainting of complex scenes.
> [2] Connelly Barnes, Eli Shechtman, Adam Finkelstein, Dan B. Goldman. PatchMatch: A Randomized Correspondence Algorithm for Structural Image Editing.

---

### Author Response · Authors · 2019-11-15
**Changes to the Paper**

Thanks to all the reviewers for their comments and suggestions. We tried to take all of them into account, we reorganized the paper accordingly and hope to provide all the required precisions.

We address below some general comments/questions raised by the reviewers and then give detailed answers for each review.

1) We have made the details of [1] clearer, as the description raised some ambiguities.
2) As suggested by review 1, we performed additional tests using [2] and [3], two SOTA unsupervised image inpainting approaches. The results are provided in the table below and have been added in Appendix E in the paper. The table shows quantitative results for  [2, 3] and our approach for the FaceForensics++ dataset with Raindrops noise. We show that the inpainting results of [2, 3] are quantitatively worse, especially for temporal quality metric FVD.

A qualitative illustration of the behavior of these models has also been added to the following site, they confirm their poor performance:
https://sites.google.com/view/unsup-video-inpaiting/.

3) Finally, we corrected the typos suggested by the reviewers.

References:
[1] Alasdair Newson, Andrés Almansa, Matthieu Fradet, Yann Gousseau, and Patrick Pérez. Video inpainting of complex scenes,
[2] Dmitry Ulyanov, Andrea Vedaldi, and Victor S. Lempitsky. Deep Image Prior.
[3] Antonio Criminisi, Patrick Pérez, and Kentaro Toyama. Region filling and object removal by exemplar-based image inpainting.

---

### Decision · Program_Chairs · 2019-12-19

**Decision:**

Reject

**Comment:**

This paper studies the problem of unsupervised inpainting occluded areas in spatiotemporal sequences and propose a GAN-based framework which is able to complete the occluded areas given the stochastic model of the occlusion process. The reviewers agree that the problem is interesting, the paper is well written, and that the proposed approach is reasonable. However, after the discussion phase the critical point raised by AnonReviewer1 remains: in principle, when applying different corruptions in each step, the model is able to see the entire video over the duration of the training. This coupled with the strong assumptions on the mask distribution makes it questionable whether the approach should be considered unsupervised. Given that the results of the supervised methods significantly outperform the unsupervised ones, this issue needs to be carefully addressed to provide a clear and convincing selling point. Hence, I will recommend rejection and encourage the authors to address the remaining issues (the answers in the rebuttal are a good starting point).